# Made in Germany: A Quality Indicator Not Only in the Automobile Industry But Also When It Comes to Skin Replacement: How an Automobile Textile Research Institute Developed a New Skin Substitute

**DOI:** 10.3390/medicina57020143

**Published:** 2021-02-05

**Authors:** Herbert Leopold Haller, Matthias Rapp, Daniel Popp, Sebastian Philipp Nischwitz, Lars Peter Kamolz

**Affiliations:** 1UKH Linz der AUVA, HLMedConsult, Zehetlandweg 7, A-4060 Leonding, Austria; 2Klinik für Orthopädie, Unfallchirurgie und Sporttraumatologie, Zentrum für Schwerbrandverletzte, Marienhospital Stuttgart, Böheimstraße 37, 70199 Stuttgart, Germany; matthias.rapp@vinzenz.de; 3Division of Hand, Plastic and Reconstructive Surgery, Department of Surgery, Medical University of Graz, A-8036 Graz, Austria; daniel.popp@medunigraz.at (D.P.); sebastian.nischwitz@medunigraz.at (S.P.N.); lars.kamolz@medunigraz.at (L.P.K.)

**Keywords:** burn, donor area, wounds, polylactide, lactormone, oxidative stress reduction, analgesia, stabilization, reduced infection

## Abstract

Successful research and development cooperation between a textile research institute, the German Federal Ministry of Education and Research via the Center for Biomaterials and Organ Substitutes, the University of Tübingen, and the Burn Center of Marienhospital, Stuttgart, Germany, led to the development of a fully synthetic resorbable temporary epidermal skin substitute for the treatment of burns, burn-like syndromes, donor areas, and chronic wounds. This article describes the demands of the product and the steps that were taken to meet these requirements. The material choice was based on the degradation and full resorption of polylactides to lactic acid and its salts. The structure and morphology of the physical, biological, and degradation properties were selected to increase the angiogenetic abilities, fibroblasts, and extracellular matrix generation. Water vapor permeability and plasticity were adapted for clinical use. The available scientific literature was screened for the use of this product. A clinical application demonstrated pain relief paired with a reduced workload, fast wound healing with a low infection rate, and good cosmetic results. A better understanding of the product’s degradation process explained the reduction in systemic oxidative stress shown in clinical investigations compared to other dressings, positively affecting wound healing time and reducing the total area requiring skin grafts. Today, the product is in clinical use in 37 countries. This article describes its development, the indications for product growth over time, and the scientific foundation of treatments.

## 1. Introduction

Stuttgart is one of Germany’s most innovative areas, as the location of the headquarters and development centers of several multinational automobile companies, including Daimler AG (Mercedes) and Porsche. Alongside a university and a technology university of applied sciences, many research companies are located there.

Among these is the German Institutes for Textile and Fiber Research Denkendorf (DITF), the largest textile research center in Europe with more than 300 scientific employees. The company’s objective is to conduct research projects that ultimately lead to developing degradable materials that do not cause any harm to the environment. The idea of using such a product as a medical wound dressing was the underlying idea behind Suprathel^®^. Developing a degradable synthetic skin substitute is a convincing approach to tackling the shortcomings of burn care and biological skin substitutes.

A joint venture between the Institute of Textile Research and Chemical Engineering Denkendorf and Stuttgart Burn Center at the Clinic for Orthopedics, Trauma Surgery and Sports Traumatology of Marienhospital, Stuttgart, Germany, was formed within the scope of the BMOZ (German Center for Biomaterials and Organ Substitute Stuttgart–Tübingen), supported by the German Federal Ministry of Education and Research, the Federal Country of Baden-Württemberg, and the University of Tübingen to develop a material “combining the advantages of biological and synthetic substitutes” [1,2].

This material was needed to advance skin substitution processes to improve burn and wound care. A transparent, permanent wound dressing was to be developed, allowing the proper assessment of the treated wound. It needed to serve as an epithelial replacement until the wound had healed completely, enabling both analgesia and the undisturbed regeneration of the damaged epithelium. The durability also had to prevent infection or iatrogenic compromise when changing the dressing. Dressing changes, if necessary, had to be as painless and straightforward as possible.

Later on, a spin-off of the institute was founded (PolyMedics Innovations GmbH) to further investigate said medical dressing and to promote its development.

This article explains the requirements of this product and reviews the developmental steps and published literature on the use of Suprathel^®^ (Polymedics Innovations GmbH. Denkendorf, Germany).

## 2. Technical Features in the Development of a Skin Substitute

### 2.1. The Choice of Materials

It was an obvious choice to use materials from components already available in the healthy human metabolism. Hydrocarbons are common and usually present in a cyclic form. As the material needs some stability over a specific time, which should reduce during degradation and resorption, polymers of lactic acid seemed to be suitable as an essential product. Polymers with a higher molecular weight degrade in a humid environment into monomers of lactic acid, buffered physiologically. It has been demonstrated that externally applied lactic acid at a lower concentration positively influences keratinocyte-derived growth factors such as VEGF (Vascular Endothelial Growth Factor [3,4]. Lactates, the salts of lactic acid, are known to scavenge superoxide radicals and to inhibit lipid peroxidation. When lactate oxidizes to pyruvate, this molecule can scavenge hydrogen peroxide and superoxide radicals as well [5,6]. Both lactic acid and lactate can enter the cells. Lactic acid, due to its small molecules and lack of polarity, being small enough to permeate through the lipid membrane, can enter cells via the monocarboxylate transporter (MCT) protein shuttle system [7]. Once inside the cell, lactate can serve as an energy source via the Cori cycle, wherein it is converted into glucose. Lactate does not cause acidosis within a physiological range [8].

Lactate dehydrogenase can alternatively convert lactate into pyruvate, which is oxidized to acetyl Co-A, producing water, carbon dioxide, and NADH (nicotinamide adenine dinucleotide plus hydrogen) in the mitochondria [9]. It can serve as an energy source and as an energy transport system [10,11].

#### 2.1.1. The Chemical Structure

The chemical structure influences the degradation time, potential toxicity, and foreign body reactions in a wound environment. The initial molecular weight and structure, as well as the chirality of polylactides, influence the speed of degradation and the degree of formation of crystalline products [12]. Schakenraad showed, in 1991, that the initial hydrolysis of amorphous poly-l-lactide might increase crystallinity and can lead to a long degradation time [13]. Therefore, the challenge was to develop a chemical compound without developing crystallinity and a foreign body reaction with a shorter resorption time.

#### 2.1.2. The Material Structure

The material had to provide a certain degree of stability, as it was intended primarily for temporary use in the place of human biological skin. This was combined with other human skin qualities such as water vapor permeability and mechanical strength, flexibility, and adherence to the wound bed.

The impact of morphology on the degradation and resorption speed was described by Taylor et al. [12]. Different structures, resulting in a different porosity, showed a different impact on angiogenesis [14,15,16,17]. Many materials with different structures and thicknesses, such as foams, fleeces, and hot melt blows, were analyzed. The angiogenic response, structural stability, and degradation had to be balanced in order to not create hypertrophic granulation tissue, to provide sufficient water vapor permeability and stability, and to preserve a moist wound environment. A balance of these properties would allow the product to be used for partial-thickness burns, supporting wound healing and the regeneration of the epithelium without generating granulation tissue. These considerations finally resulted in the pore size and structure used in Suprathel^®^ (Polymedics Innovations GmbH, Denkendorf, Germany).

## 3. Results of the Development Process

### 3.1. Material Properties

This development process resulted in a specific microporous structure for Suprathel^®^. This microporous structure provides an increased moisture vapor permeability at the beginning of the treatment. Induced by fibrine and the outer dressing, the water vapor permeability of Suprathel^®^ is reduced over the time to that of healthy human skin.

Suprathel^®^ consists of DL-lactide (>70%), trimethylene carbonate, and ε-caprolactone. A unique processing technique creates a porous membrane with a nearly symmetrical cross-section, with an interconnected pore structure and varying pore sizes between 2 and 50 μm. The initial porosity of the material is >80%. Due to its plasticity, it adapts to the wound bed at body temperature. It promptly adheres to a fresh wound. The membrane has a thickness of 70–150 μm and an elongation potential between 100% and 250% [18]. Storage at room temperature is possible for three years without the loss of quality. Degradation and loss of stability occur within four weeks in vitro; in vivo, it is stabilized by fibrin, and the separation layer and remnants are removed after wound healing. Examples for the use of Suprathel^®^ are shown in Figure 1 and Figure 2a,b.

### 3.2. Medical Aims of Suprathel Development

Table 1 gives an overview on the medical key points of the development of Suprathel^®^. The effects are described in detail in the following paragraphs.

### 3.3. Detailed Results of Development

#### 3.3.1. Basic Mechanism

Suprathel^®^ offers positive effects on wound healing. These include the barrier effect, the effect on dermal tissue, the effect on epidermal tissue, and systemic effects. The barrier effect provides bacterial tightness and reduces fluid and energy loss [1,19]. The effect of lactate has been described above. It works as a radical scavenger and energy source for cells. Lactate simulates hypoxic conditions for cells, while oxygen tension is normal [11], supporting the metabolism and wound healing in dermal tissue. It increases VEGF [20], the number of vessel sprouts and functional vessel density, red blood cell velocity, as well as interconnections in capillaries. It supports fibroblast migration and extracellular matrix generation [21] and collagen synthesis [22]. Lactate’s effect on TGF-β (Transforming Growth Factor β) generation is limited to two weeks, preventing myofibroblast generation but supporting early wound healing [23,24,25]. In epidermal tissue, it supports keratinocyte activation by redifferentiation and forming filopodia, enabling keratinocyte migration and wound closure under lactate-induced TGF-β [26]. Cleavage during wound closure by decorin limits this effect [27]. Systemic effects support burn wound healing by influencing the inflammatory response, as demonstrated in children with a mean TBSA (total burned surface area) burn of 39.95% [28].

#### 3.3.2. Pain Reduction

Suprathel^®^ has been shown to significantly reduce pain when applied to fresh wounds. This effect is nearly unique to Suprathel^®^ and is only comparable to dressings with an incorporated analgesic and antiphlogistic medication such as Ibuprofen [29]. One reason for this might be the radical scavenging ability of the free radicals released after trauma by keratinocytes and other cells [28]. ROS (reactive oxygen species) induces the synthesis of PGE2 (Prostaglandin E2) [30]. A specific PGE2 receptor, EP1, is responsible for pain perception. EP1-knockout mice only show 50% of the pain perception of wild-type mice. The reduction in free radicals results in reduced pain triggers, as demonstrated by Stock et al. [31]. Reduced PGE2 also influences local edema formation as a component of inflammation [31], as demonstrated by Kaartinen et al. [32]. The reduction in oxidative stress and pro-inflammatory cytokines might significantly influence the reduction in pain [28,33].

Another reason might be the protective effect on wound desiccation, known from other materials [34], hindering pain from free nerve endings [35]. This effect alone is not sufficient to explain the degree of pain reduction in Suprathel^®^. When compared directly to other products, Suprathel^®^ shows superiority in hindering desiccation [32]. Suprathel^®^ can significantly reduce resting and procedural pain due to its structure, as well as to inner dressing changes being omitted [36,37,38]. As the Suprathel^®^ skin template covered with the separation layer remains adherent until the wound is healed, it protects the wound from irritation and disruption during dressing changes. The degradation of the absorbent layer does not trigger additional pain.

##### The Practical Meaning of Pain Reduction

Based on its physiological effect, several randomized controlled trials have demonstrated the excellence of Suprathel^®^ in pain reduction. Significant superiority was found when compared to Jelonet [39], Mepilex [29,32], and Omiderm [40]. Pain reduction, which was in the range of 30–63%, was statistically significant in all of these studies. Biatain Ibu showed a faster but weaker effect than Suprathel^®^ but had to be changed after three days to preserve the pain-reducing ability [29]. To date, more than 20 studies, including prospective, retrospective, and case reports, have confirmed the excellent effect on pain reduction in partial-thickness burns [38,41,42,43,44,45] and donor sites, radiation injuries [46], and burn-like syndromes [47,48,49], as well as successful use in high-risk and complicated patients [50,51] and cauterizations [52]. Some publications have even demonstrated a significant reduction in opioids [37,45,51,53,54].

The other positive side effects include pain reduction, which has several positive side effects. This enables the early return of function and early mobilization, as well as a lower need for sedation, preparation, and surveillance before, during, and after procedures. Furthermore, it reduces the side effects of opioid use, such as obstipation, dizziness, loss of appetite, and hypotension, which might be compensated for by more fluid application linked with edema and fluid-creep, triggering dependency on opioids [55,56]. Short et al. [52] and others [44] have reported that it also enables outpatient care.

The reduction in opioids means less of a sedation effect, which facilitates early extubating and prevents the complications that come with longer periods of artificial ventilation, as demonstrated in the Burn Repository 2017 of the ABA (American Burn Association) [57] by reductions in fluid retention and acute kidney injury [58,59,60,61]. The goal of opioid reduction is to prevent long-term opioid use (opioid dependency) [62].

#### 3.3.3. The Effect of Treatment with Suprathel^®^ on Workload and Stress to Both Patients and Medical Staff

The basic principle for Suprathel^®^ treatment is the initial application of Suprathel^®^ to the wound. The outer dressing typically consists of a separation layer (i.e., fatty gauze), followed by an absorbent dressing and a compression layer. Only the outer dressing is replaced during dressing changes.

This results in a reduced frequency of dressing changes and a reduced overall time necessary for each dressing change, since the inner dressing remains attached to the wound bed. Kukko et al. found a reduction in dressing changes to 42% compared to SSD silver sulfa diazine (SSD) [63], while Markl et al. found a reduction from a mean of 4.0 in Mepitel to 0.3 in Suprathel (7.5%). The time needed for dressing changes was Ø2.25 min for Suprathel^®^ and Ø10.58 min for Mepitel^®^.

#### 3.3.4. Effects of Suprathel^®^ on Water Vapor Permeability and Fluid Management on the Patient

Suprathel^®^ reduces bleeding and is permeable for water and serum during the initial treatment and causes less fluid secretion than other products [32]. It works well on donor sites as it does not form a “cement-like clot” like other products (e.g., Biobrane) [64]. Hence, donor sites are an excellent indication for Suprathel^®^.

As several case reports show, in products such as Biobrane^®^ [65,66], Integra^®^ [67], and Opsite^®^ [68], or even skin grafts [69], toxic shock syndrome can be a severe side effect that is likely related to fluid retention underneath the dressing. There are no reports of toxic shock syndrome with the use of Suprathel^®^. This might be due to the increased water vapor permeability.

Suprathel^®^ does not support hematoma formation or fluid retention at the beginning of treatment, and therefore observation is less frequently necessary. This is in contrast to several other materials, which have to be closely observed for these complications [70]. The water vapor permeability of Suprathel^®^ at the beginning of the treatment is below 50% of a partial thickness burn in vitro. This reduces even further to below 25% within 20 days [71]. In vivo, it links with fibrin and cells from the wound, such as red blood cells, thrombocytes, and leukocytes, reducing permeability within four to five days to nearly 0%. After this time, the dressing stays dry, and further dressing changes are only necessary for on-demand or hygienic reasons.

#### 3.3.5. Effects of Suprathel^®^ on the Energy Balance

Water vapor permeability is an essential component of energy balance. Loss of water and serum over the wound causes a wet surface, resulting in cooling by evaporation. This results in a drop in body temperature, and hence energy is lost in an attempt to maintain body temperature. Occlusive dressings can substantially reduce energy requirements [72,73]. The semi-occlusive properties of Suprathel^®^ at the beginning of treatment, turning into a nearly occlusive treatment over the early treatment course, might reduce energy requirements and the inflammatory response and therefore improve the metabolic situation in favor of a better wound healing situation [74,75]. Demircan demonstrated a substantial reduction in healing time after Suprathel^®^ treatment, possibly reducing oxidative stress and attenuating energy loss. This attenuation of energy balance might also lead to a reduction in areas that require skin grafts, as demonstrated by Uhlig et al. [76], Schriek et al. [77], and Blome-Eberwein et al. [78].

#### 3.3.6. Reduction in the Oxidative Stress and the Systemic Inflammatory Response

Gürünlüoglu et al. investigated the effect of Suprathel^®^ on the systemic inflammatory response. They found a positive impact on the total antioxidant and reduced total oxidant capacity compared to hydrofiber (HF) Ag at a highly statistically significant level [28]. Wound healing was also significantly faster in the Suprathel^®^ group. They were the first to demonstrate the systemic effects of wound dressings on oxidative stress in humans.

The same group also investigated the impact of Suprathel^®^ on cytokine levels. IL-6 (Interleukin 6) and TNF-α (Tumor Necrosis Factor α) were significantly lower in the Suprathel^®^ group than in HF Ag, and TGF-β was significantly higher throughout the two weeks and reduced to nearly normal levels after the third week [25].

Gürünlüoglu et al. also demonstrated the impact of oxidative stress reduction on skin quality using Suprathel^®^ by measuring telomerase activity in healed skin after Suprathel^®^ treatment compared to HF Ag. This study demonstrated higher telomerase activity and a higher cell count, with higher skin viscoelasticity, in the Suprathel^®^ group, concluding a better result in regard to skin quality after Suprathel^®^ treatment [79].

#### 3.3.7. The Practical Meaning of the Reduction in Oxidative Stress

Severe thermal injury induces a pathophysiological response that affects most of the body’s organs, including the liver, heart, lungs, and skeletal muscle, and leads to inflammation and hypermetabolism as a hallmark of post-burn damage. Oxidative stress has been implicated as a critical component in developing an inflammatory and hypermetabolic response after a burn [80]. Mitochondria are a primary site of free oxygen radical metabolism. In burn victims, it has been found that a burn-related cardiac dysfunction involves mitochondria under oxidative stress [81]. Hypermetabolism and Hyperkatabolism have been found to accompany mitochondrial stress. Burn trauma alters mitochondrial defenses against free oxygen radicals, which can be positively influenced by antioxidants [81]. Suprathel^®^ applied topically systemically reduces oxidative stress and might decrease organ damage as a consequence [28].

The semi-occlusive properties and a lower inflammatory response by reducing oxidative stress can explain the reduced need for fluid and the patient’s early stabilization with Suprathel^®^ treatment [30,74,82]. Free radicals cause a “shedding” of the mostly hyaluronic acid-based glycocalyx of the capillaries and vessels, causing a higher permeability for fluids and the formation of edema, which can be reduced by the radical scavenging abilities of Suprathel^®^. Systemic stabilization under Suprathel^®^ treatment can occur, as Rubenbauer et al. described in a case of epidermal necrolysis treated with a silver product prior to Suprathel^®^. The patient developed fluid retention with massive potassium needs and required intubation and ventilation. After changing the treatment to Suprathel^®^, the patient stabilized quickly, could be extubated on day one after Suprathel^®^ treatment, and was shortly thereafter able to be discharged from the ICU (intensive care unit) [74]. Short described similar cases [51].

A significant reduction in the need for fluids with Suprathel^®^ was reported by Hentschel et al. when comparing Suprathel^®^ to the standard of care (SOC) [81]. The reduced need for fluids with less edema formation might aid in reducing complications such as compartment syndrome, organ failure, and reduced graft take.

Burn wound conversion is the progression of tissue damage over the first days after a burn, resulting in the deepening of the burn. Clinically, areas primarily evaluated as healing spontaneously convert to areas that need skin grafts.

In two retrospective studies, a reduction in the need for grafting in Suprathel^®^-treated cases versus SOC indicated a reduction in burn wound conversion. The degree of reduction ranged from 27% (historical SOC) to 7% and 2.1% [36,77]. Prospective studies have shown reduced systemic oxidative stress [28] and inflammatory responses [25], possibly as critical causes for reducing burn wound conversion. Additionally, Suprathel^®^ positively affects telomeric kinetics, a significant event in oxidative stress regulation [79]. Apoptosis, combined with connexin’s upregulation, seems to be part of burn wound progression [83]. Anti-apoptotic measures such as reducing connexin by addition of poly-L-lactide, as a component of Suprathel^®^, reduces apoptosis and supports faster wound healing [84]. All of these components support the positive effect of Suprathel^®^ on burn wound conversion.

As there are many influencing components of burn wound conversion, Suprathel^®^ has been shown to potentially reduce some of them. It hinders desiccation, reduces free oxygen radicals and lipid peroxidation, reduces the inflammatory response, and undergoes sympathetic activation by pain, followed by minder perfusion. It supports cells with energy by the lactate shuttle [85], and releases wound healing cytokines [33]. Reduced burn wound conversion means a reduction in areas to be grafted and reduces donor areas, meaning a faster reduction in the total wound area and positively influencing the course of healing. As demonstrated by Blome-Eberwein et al., Suprathel^®^ can reduce the number of patients with partial-thickness burns to be grafted from 20% previously to 0% of all cases, compared to historical SOC, after three weeks. Schriek et al. showed that this reduced from 30% in the historical group to 7–9% after 11 days [77,78].

### 3.4. Studies and Reports on the Use of Suprathel^®^ for Different Indications

Suprathel^®^ is a versatile dressing in burns treatment and can be used for most indications during treatment. More than 160 reports, studies, and abstracts have been published for various indications.

#### 3.4.1. Donor Areas

Donor areas are often neglected in the treatment of skin defects. Particularly in older people, non-healing donor sites can cause long-lasting morbidity and burden [86]. Suprathel^®^ has been successfully used for donor areas, and many studies have shown excellent results. Pain reduction, reduced workload, very low infection rates, and fast healing times are the main advantages [32,54,71,87]. Berg presented a case with repetitive harvesting 12 times from the scalp without causing alopecia [88]. Healing time can be expected in the range of seven and eight days in children [42,87,88] and from nine days (clinical experience) to 13 days in adults [29].

#### 3.4.2. Superficial and Partial-Thickness Burns

Superficial and partial-thickness burns covered with Suprathel^®^, especially in more extensive burns, provide a comfortable and nearly painless treatment, with reduced stress for patients and reduced stress and workload for medical staff [89]. It can be combined with the tangential excision or enzymatic debridement in partial-thickness burns, and given the presence of a sufficient number of epidermal remnants, healing without complications can be expected in deeper partial-thickness burns, it can reduce the need for grafting [90,91,92,93] and can provide better cosmetic results than in mesh-grafted areas. This has been confirmed by validated scar scales (i.e., POSAS (Patient and Observer Scar Assessment Scale) and VSS (Vancouver Scar Scale) six months after the burn. The cosmetic result measured on a five-point scale was significantly better after one month with Suprathel^®^, and patients were more satisfied than with Mepilex^®^ [32].

#### 3.4.3. Partial-Thickness and Small Third-Degree Burns

In burns that are mostly partial thickness with small areas of only third-degree burns, treatment with Suprathel^®^ is particularly useful. The whole wound can be treated with Suprathel^®^. The majority of the wound will heal spontaneously, and only the remaining small areas that do not heal spontaneously within a defined range of time will have to be excised and grafted. This leads to an overall better cosmetic outcome with fewer surgical procedures [76,93,94]. This technique can also be applied in more extensive mixed burns. In areas that will not heal spontaneously, Suprathel^®^ will not be adherent, and the need for grafting will be clear after 10–14 days [76]. During this time, the patient can be stabilized and/or transported. Thus, Suprathel^®^ is the ideal dressing for burn treatment for mass casualties at a lower level of care or under combat conditions. After incomplete debridement and disinfection and covering with Suprathel^®^, the patient can be transported, and excision of necrotic tissue can be done in a delayed manner under controlled infection conditions [95]. As mentioned above, the main advantages are pain reduction, attenuated fluid balance, and protection from infection.

#### 3.4.4. Full-Thickness Burns

After the complete excision of a burn wound, Suprathel^®^ can be used as a temporary dressing, reducing pain and workload with the advantage of avoiding skin staples, as only the external dressing needs to be changed, which reduces blood and fluid loss [76,96,97].

#### 3.4.5. Suprathel^®^ and Cells

Suprathel^®^ has shown excellent adhesion over and proliferation of keratinocytes and fibroblasts [98,99]. The results of covering sprayed keratinocytes were first published in 2012 [100,101,102]. The healing time of Suprathel^®^-covered sprayed cultured and non-cultured keratinocytes was better or equivalent to the results with other dressings published in the literature [101,103]. After application to deep partial-thickness burns, the mean healing time was, on average, eight days after the clinical evaluation of the wound depth and 10 days after laser doppler imaging (LDI). In comparison, the wounds healed, on average, three days faster than in other studies. Furthermore, Suprathel^®^ also protected these wounds with applied cells from infection [104,105].

#### 3.4.6. Suprathel^®^ and the Meek Technique

After removing a Meek nylon pleated wound coverage after 10–14 days, Suprathel^®^ is a good option for covering the incompletely healed wound areas. Suprathel^®^ can be left on the confluent Meek Islands until the wound is fully healed, allowing for more secure epithelialization. This procedure leads to faster healing and a reduction in the rate of wound infections [106].

#### 3.4.7. Suprathel^®^ after Enzymatic Debridement

Suprathel^®^ has been suggested as a dressing after enzymatic debridement (ED) in several consensus meetings, mainly because of the reduced need for dressing changes and its wound-healing abilities. In the European consensus update, it was declared as appropriate, and in the Spanish consensus [107,108,109], it was declared as a most appropriate dressing material after E.D. in different publications [107,108,109,110,111,112]. Due to the increased effusion after E.D., a prolonged after soaking period of 5–8 h is indicated [112]. In a retrospective study assessing healing time after E.D., Suprathel^®^ showed a shorter healing time overall than polyurethane foam and cream application, especially in burns with a higher Baux Index, offering all the known advantages of Suprathel^®^ [113,114].

#### 3.4.8. Suprathel^®^ in Burn-Like Syndromes

Excellent results, especially in high-risk toxic epidermal necrolysis (TEN) cases, have been described, including the youngest case ever published before 2006 [115]. The main factors might be reducing the overall mortality due to insufficient wound care, linked with pain reduction, reduced stress, the prevention of life-threatening infections, and the stabilization of the patient [115,116,117,118,119,120]. It has been shown to be superior to an allograft. Successful case reports have been published as well on staphylococcal scalded skin syndrome [49], phototoxic plant burns [121,122,123], frostbites [124,125], cotton wool babies [126], and aplasia cutis [127].

#### 3.4.9. Suprathel^®^ in Radiation Injuries

Rothenberger et al. described its successful use in exfoliative dermatitis [46]. Radiation therapy could be continued from 40 Grey to 64 Grey while the wound was able to heal. Immediate pain relief was a positive side effect. Similar cases are currently being peer reviewed.

#### 3.4.10. Suprathel^®^ in Ulcus Cruris

Sari et al. described the positive effect of Suprathel^CW^ in treating chronic wounds in venous, arterial, diabetic, and mixed ulcers. The use of Suprathel^CW^ resulted in a reduction in size or healing of the ulcer. Pain levels were significantly reduced, the skin’s inflammatory reaction was significantly reduced, and the wound secretion decreased [128].

### 3.5. Cost-Effectiveness

When describing costs, material costs and total treatment costs must be taken into account. For Suprathel^®^, the lower total costs were described by Schwarze et al. [40], Everett et al. [37], and Fischer et al. [50].

## 4. Limitations

In practical use, instructions have to be followed. It is necessary to overlap Suprathel^®^ over the wound margins and to overlap between sheets. A separation layer and some compression must be applied to prevent adherence to the external absorbent layers and the dislocation of the Suprathel^®^.

In dressing changes, Suprathel^®^ stays in place, as does the separation layer. Only the absorbent layer and the compressions are changed. Not to do so, causes complications, as described by Rashaan et al. [43]. Experience in the use of Suprathel^®^ in full-thickness burns and ulcers is limited and must be expanded. Costs are sometimes described as a limiting aspect. Although the material may be more expensive than other treatment modalities, different authors have described its cost-effectiveness when considering the total treatment costs.

## 5. Conclusions

The positive cooperation of a tech company with research institutes and clinical departments, with public funding support and a spin-off, proven to be the source of success for developing and further improving a medical product, which was primarily not in the company’s scope.

A fully synthetic and fully resorbable epidermal template based on lactic acid, a physiological and essential product of the human metabolism, opened a new product line and eliminated complicated and laborious production steps. Biocompatibility has been achieved before in other products by the genetic modification of animal-derived tissue [129], which remains limiting in terms of viral, bacterial, and prion safety. By using physiological components only in Suprathel^®^ (i.e., lactic acid), these safety limitations could be eliminated [130].

After the initial regulatory studies, scientific curiosity expanded and later helped find essential wound healing mechanisms relevant to the product. Physical properties such as the adaptation of water vapor permeability over time supported by the body wound healing interactions made a new dressing and treatment scheme possible, reducing complications and the need for inner dressing changes, in contrast to other products [32,130]. A new understanding of lactate and its mechanism positively impacted the local and systemic inflammatory response and (hyper-) metabolism and cell regeneration compared to other products [28,78]. Lactate’s effects on particular pain-related receptors could explain the strong pain-reducing effect [131].

## Figures and Tables

**Figure 1 medicina-57-00143-f001:**
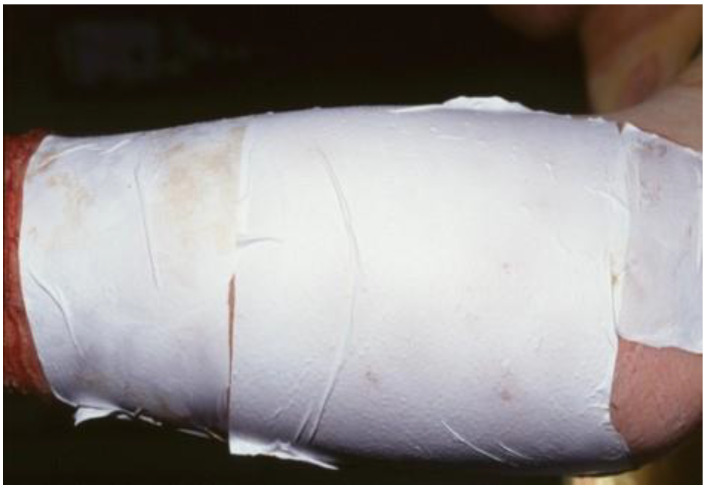
Shows the first application of Suprathel^®^ in 1999 in a donor site (courtesy of Dr. Rapp).

**Figure 2 medicina-57-00143-f002:**
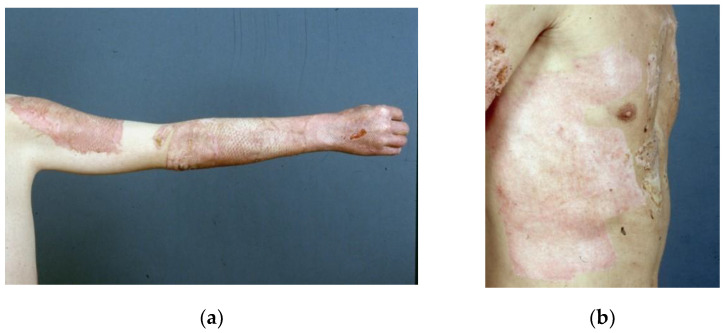
(**a**) Shows healed areas after the application of Suprathel on a mesh-grafted area; and (**b**) shows a healed donor area after Suprathel application on day 13 post op. (courtesy of Dr. Rapp).

**Table 1 medicina-57-00143-t001:** Medical and handling key points of the development of Suprathel^®^.

Pain reduction;
Reduction in oxidative stress and the systemic inflammatory response;
Reduction in the need for skin transplantations and consecutive donor wounds;
Support for wound healing and more efficient healing;
Low infection rate;
Good cosmetic results and scar quality;
Reduction in workload;
Economic efficiency.

## Data Availability

No new data were created or analyzed in this study. Data sharing is not applicable to this article.

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
