# Peer review of "Made in Germany: A Quality Indicator Not Only in the Automobile Industry But Also When It Comes to Skin Replacement: How an Automobile Textile Research Institute Developed a New Skin Substitute"

_medicina, 2021, doi:10.3390/medicina57020143_

Round 1
Reviewer 1 Report
The authors describe the development of an epidermal skin substitute (Suprathel) as result of a cooperation between industry, research institutes and clinicians. By means of a large reference list of peer reviewed publications and presentations an excellent overview about the effectiveness and the results of treatment of different skin diseases, in the first instance of burn wounds, is given. The whole manuscript is an excellent overall review about a product and its 20-year-history which meanwhile has a high significance in burn treatment.
In my opinion only some citations in the reference list need revision - e.g. 78=98; 79 ; 95 (abstract 096); 96.
Author Response
Dear Reviewer 1,
Thank you for the time and effort you spent on reviewing the manuscript.
Line 358: we eliminated the duplicate reference
Line 645: we corrected the duplicate authors
Line 684: we corrected the spelling of anstract to abstract
Line 685: we corrected the duplicate authors
Thank you for the support and positive evaluation of our manuscript
The authors
Reviewer 2 Report
The reviewer commends the authors on their extensive review about the development and clinical properties of Suprathel®. Despite its regular use by both Plastic and Burn Surgeons worldwide, I can only assume that a small fraction of these users is aware that Suprathel® was originally developed in cooperation with the German automobile industry. This is a very interesting fact that would fit nicely within the Journal’s special issue entitled, “A History of Burn Care.”
I have only minor suggestions:
- Line 4-5: From “How a textile research institute usually working for car development developed a new skin substitute” to “How an automobile textile research institute developed a new skin substitute.”
- Line 27: Change: “reducing the areas to be grafted” to: “reducing the total area requiring skin grafts.”
- The sentence in Line 56/57: “The durability also had to prevent infection or iatrogenic compromise when changing the dressing was changed.” This sentence does not make sense. Please revise.
- Line 63: “2. Technical Features in the Development of a Skin Substitute”. The reviewer recommends making this a new subchapter in bold letters.
- For line 113/114 the reviewer suggests changing: “Induced by fibrine and the outer dressing, the water vapor permeability of Suprathel® reduces to below that of healthy human skin.” To: “Induced by fibrine and the outer dressing, the water vapor permeability of Suprathel® is reduced over time to below that of healthy human skin.”
- Line 168-170: “One reason for this might be the radical scavenging ability of the free radicals released after trauma by keratinocytes and other cells [30]. Is there a reason for this to be in bold font?
- Line 423-425: The reviewer suggests changing the sentence from: “A separation layer has to be applied, hindering the external absorbent layers’ adherence, and some compression must be applied.” To: “A separation layer and some compression must be applied to prevent adherence to the external absorbent layers and dislocation of the Suprathel®, ”
Author Response
Dear Reviewer 2,
Thank you for your time and effort spent on the intensive study of the paper and your suggestions.
Line 4-5: We followed your suggestion and changed the title part to “How an automobile textile research institute developed a new skin substitute.”
Line 27: We changed the wording “reducing the areas to be grafted” to “ reducing the total area requiring skin grafts.”
Line 63 is Line 62 in the paper: “2. Technical Features in the Development of a Skin Substitute”. The reviewer recommends making this a new subchapter in bold letters. We made a subchapter in bold letters, following your suggestion.
Line 113/114: (112/113) the reviewer suggests changing: “Induced by fibrine and the outer dressing, the water vapor permeability of Suprathel® reduces to below that of healthy human skin.” To: “Induced by fibrine and the outer dressing, the water vapor permeability of Suprathel® is reduced over time to below that of healthy human skin.” We followed the suggestion
Line 168-170: “One reason for this might be the radical scavenging ability of the free radicals released after trauma by keratinocytes and other cells [30]. There is no reason, and we changed it to the normal script.
Line 423-425: The reviewer suggests changing the sentence from: “A separation layer has to be applied, hindering the external absorbent layers’ adherence, and some compression must be applied.” To: “A separation layer and some compression must be applied to prevent adherence to the external absorbent layers and dislocation of the Suprathel®, ”
We followed this suggestion.
Again, thank you for your positive support!
The authors